# DIFFERENTIABLE PROMPT MAKES PRE-TRAINED LANGUAGE MODELS BETTER FEW-SHOT LEARNERS

**Ningyu Zhang**[1,2,3*]   **Luoqiu Li**[1,3*]   **Xiang Chen**[1,3]   **Shumin Deng**[1,3]   **Zhen Bi**[2,3]
**Chuanqi Tan**[5]   **Fei Huang**[5]   **Huajun Chen**[1,3,4†]
[1]College of Computer Science and Technology, Zhejiang University
[2]School of Software Technology, Zhejiang University
[3]Alibaba-Zhejiang University Joint Research Institute of Frontier Technologies
[4]Hangzhou Innovation Center, Zhejiang University
[5]Alibaba Group
`{zhangningyu,3160102409,xiang_chen,231sm,bizhen_zju}@zju.edu.cn,`
`{chuanqi.tcq,f.huang}@alibaba-inc.com`

## ABSTRACT

Large-scale pre-trained language models have contributed significantly to natural language processing by demonstrating remarkable abilities as few-shot learners. However, their effectiveness depends mainly on scaling the model parameters and prompt design, hindering their implementation in most real-world applications. This study proposes a novel pluggable, extensible, and efficient approach named DifferentiAble pRompT (DART), which can convert small language models into better few-shot learners. The main principle behind this approach involves reformulating potential natural language processing tasks into the task of a pre-trained language model and differentially optimizing the prompt template as well as the target label with backpropagation. Furthermore, the proposed approach can be: (i) Plugged to any pre-trained language models; (ii) Extended to widespread classification tasks. A comprehensive evaluation of standard NLP tasks demonstrates that the proposed approach achieves a better few-shot performance[1].

## 1 INTRODUCTION

The pre-train—fine-tune paradigm has become the de facto standard for natural language processing (NLP), and has achieved excellent results in several benchmarks (Devlin et al., 2019; Liu et al., 2019; Lewis et al., 2020; Dong et al., 2019; Bao et al., 2020a). The success of these pioneers seems to suggest that large-scale pre-trained models are always nothing short of a panacea for boosting machine intelligence. However, supervised fine-tuning is still prone to labeled data in practice and faces unignorable challenges owing to the variations of domains, language, and tasks. These drawbacks lead to the research of an important technique, *few-shot learning*, which can significantly improve the learning capabilities of machine intelligence and practical adaptive applications by accessing only a small number of labeled examples.

The GPT-3 model, introduced by Brown et al. (2020), exhibits impressive few-shot learning capabilities. Given a natural language prompt and 16 labeled samples as demonstrations in the contextual input, GPT-3 achieves 80% of the SOTA results. However, GPT-3 is a fully dense transformer model with 175B parameters, which makes it challenging to deploy in most real-world applications.

Recently, an emerging fine-tuning methodology has arisen to equip smaller language models (LMs) with few-shot capabilities: adapting the pre-trained LM directly as a predictor through completion of a cloze task (Schick & Schütze (2021; 2020); Gao et al. (2020); Liu et al. (2021c)), which treats the downstream task as a (masked) language modeling problem. These prompts can be used in fine-tuning to provide the classifier with additional task information, especially in the low-data regime.

---

[*]Equal contribution and shared co-first authorship.
[†]Corresponding author.
[1]Code is available in `https://github.com/zjunlp/DART`.

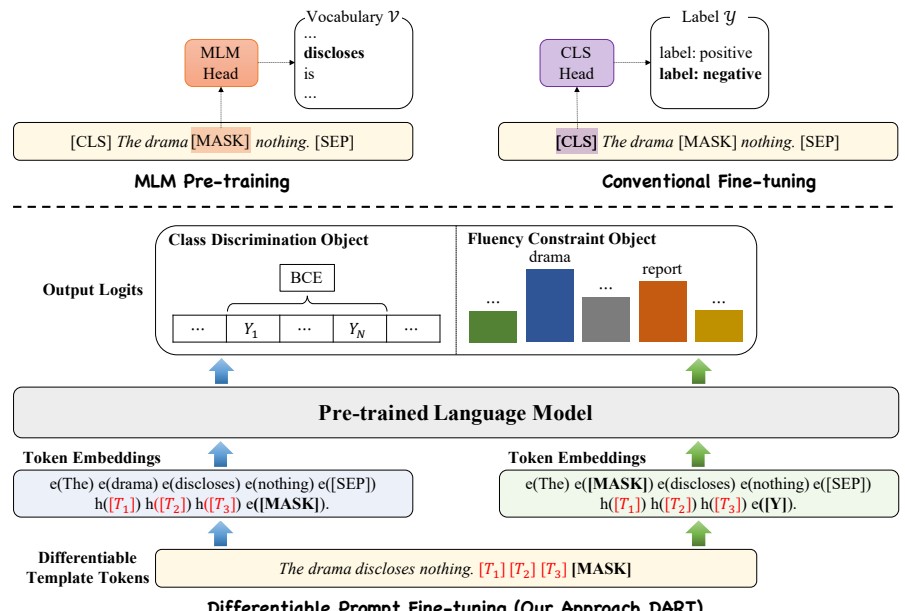

Figure 1: The architecture of **D**ifferenti**A**ble p**R**omp**T** (**DART**) model comparing with *MLM pre-training* and *conventional fine-tuning*, where $T_i$ and $Y_i$ are unused or special tokens in the vocabulary. We leverage a few parameters within the language model as the template and label tokens and optimize them via backpropagation without introducing additional parameters apart from the model.

Notably, Scao & Rush (2021) observe that prompting can often compensate for hundreds of data points on average across multiple classification tasks. However, determining the appropriate prompts requires domain expertise, and handcrafting a high-performing prompt often requires impractically large validation sets (Perez et al. (2021)). Recent studies (Lu et al. (2021); Zhao et al. (2021)) have reported that the manual prompt format can be sub-optimal, which would result in the accuracy varying from random guess performance to near the state-of-the-art. Therefore, previous approaches have attempted to search for discrete prompt tokens automatically. However, it is non-trivial for widespread classification tasks to obtain an optimized prompt template and target label token. For example, specific classification tasks such as relation extraction with the label of *alternate_name* and *country_of_birth* cannot specify a single label token in the vocabulary.

In this paper, we propose a novel **D**ifferenti**A**ble p**R**omp**T** (**DART**) fine-tuning approach, which is model-agnostic, parameter-efficient. As illustrated in Figure 1, the key idea is to leverage a few parameters (unused tokens) in the language model, which serve as the template and label tokens, and to optimize them in the continuous space using backpropagation. Subsequently, we introduce differentiable prompt learning to obtain optimized prompt templates as well as labels. Since fine-tuning with limited samples can be affected by instability (Dodge et al. (2020); Zhang et al. (2021)), we propose an optimization algorithm to jointly learning templates as well as labels. We further introduce an auxiliary fluency constraint object to ensure the association among the prompt embeddings.

We conduct extensive experiments on 15 NLP datasets. With only a few training samples across all the tasks, our approach (DART) can obtain a better performance. Notably, absolute performance improvement of up to 23.28%, over the conventional fine-tuning, is obtained on average in the setting of $K = 8$ (and 1.55% for fully supervised settings) on relation extraction datasets with complex label semantics. Our approach can be applied to real-world classification tasks without the high cost of collecting and annotating a large amount of data. The main contributions of this study are as follows:

- We propose a new simple framework for few-shot learning, which is pluggable, extensible, and efficient. To the best of our knowledge, optimizing label tokens in continuous space is also a new branch of research that has not been explored in language model prompting.

- A systematic evaluation of 15 NLP tasks shows that the simple-yet-effective method contributes towards improvements across all these tasks. Remarkably, given only 8 labeled samples per class, our proposed approach can achieve 90% performance of the SOTA results (full dataset).

## 2  RELATED WORK

**Language Model Prompting.**  The language model prompting has emerged with the introduction of GPT-3 (Brown et al. (2020)), which demonstrates excellent few-shot performance (Liu et al. (2021b)). However, GPT-3 is not designed for fine-tuning; it mainly relies on the handcraft prompt (in-context learning (Liu et al. (2021a); Zhao et al. (2021); Ding et al. (2021); Min et al. (2021))). Thus, recent studies (Qin & Eisner (2021); Hambardzumyan et al. (2021); Chen et al. (2021)) conducted in this field have been focused on automatically searching the prompts. Schick & Schütze (2021; 2020) propose the PET, which reformulates the NLP tasks as cloze-style questions and performs gradient-based fine-tuning. Tam et al. (2021) improve the PET with a denser supervision object during fine-tuning. Shin et al. (2020) propose the AUTOPROMPT to create prompts for a diverse set of tasks based on a gradient-guided search. Han et al. (2021) propose an approach called PTR, which leverages logic rules to construct prompts with sub-prompts for many-class text classification. Wang et al. (2021) reformulate potential NLP task into an entailment one, and then fine-tune the model with few-shot samples. Hu et al. (2021) propose an approach to incorporate external knowledge graph into the verbalizer with calibration. Additionally, Gao et al. (2020) present LM-BFF—better few-shot fine-tuning of language models, which leverages T5 (Raffel et al. (2020)) to generate templates and search label tokens in the vocabulary. However, the utilization of the generative model and the label search with validation is computation-intensive. Moreover, the prompt search over discrete space is sub-optimal due to the continuous nature of neural networks.

To overcome these limitations, Liu et al. (2021c) propose P-tuning, which employs trainable continuous prompt embeddings learned by an LSTM. Zhong et al. (2021) propose an effective continuous method called OPTIPROMPT to optimize prompts for factual probing. Liu et al. (2021c) propose prefix-tuning, which keeps language model parameters frozen but optimizes a small continuous task-specific vector for natural language generation tasks. Lester et al. (2021) propose a mechanism for learning "soft prompts" to condition frozen language models to perform downstream tasks. However, these approaches still have to optimize the external parameters (e.g., LSTM in P-tuning) and are prone to complex label space.

Conversely, this study aims to develop a novel few-shot learning framework based on pre-trained language models which can reduce the prompt engineering (including templates and labels) and external parameter optimization. Furthermore, the proposed approach only leverages the noninvasive modification of the model, which can be plugged into any pre-trained language model and extended to the widespread classification task.

**Few-shot Learning.**  Few-shot learning can significantly improve the learning capabilities for machine intelligence and practical adaptive applications by accessing only a small number of labeled examples (Zhang et al. (2020)). The proposed approach corresponds to the other few-shot NLP methods, including: (1) Meta-learning (Yu et al. (2018); Bao et al. (2020b); Bansal et al. (2020); Deng et al. (2020b;a); Yu et al. (2020)), in which the quantities of the auxiliary tasks are optimized. (2) Intermediate training (Phang et al. (2018); Yin et al. (2020)), which supplements the pre-trained LMs with further training on the data-rich supervised tasks. (3) Semi-supervised learning (Miyato et al. (2017); Xie et al. (2020)), which leverages unlabeled samples. The proposed approach focuses on a more realistic few-shot setting (the number of labeled instances per class can be any variable).

## 3  BACKGROUND

Let $X_{\text{in}} = \{x_1, x_2, ..., x_L\}$ be a sentence, where $x_i$ is the $i^{th}$ token in the input sentence and $L$ is the number of tokens. Specifically, $X_{\text{in}}$ is converted to a fixed token sequence $\tilde{X}_{\text{in}}$ and then mapped to a sequence of hidden vectors $\{\mathbf{h}_k \in \mathbb{R}^d\}$. Given the input sequence, $\tilde{X}_{\text{in}} = \text{[CLS]}\, X_{\text{in}}\, \text{[SEP]}$, the conventional fine-tuning approaches leverage a generic head layer over [CLS] embeddings (e.g., an MLP layer) to predict an output class. For the prompt-based method, a task-specific pattern string

(template $\mathcal{T}$) is designed to coax the model into producing a textual output corresponding to a given class (label token $\mathcal{M}(Y)$)—we refer to these two things together as a prompt. Specifically, $X_{\text{prompt}}$ containing one [MASK] token is directly tasked with the MLM input as:

$$X_{\text{prompt}} = [\text{CLS}]\,X_{\text{in}}\,[\text{SEP}]\,\mathcal{T}\,[\text{SEP}] \tag{1}$$

When the prompt is fed into the MLM, the model can obtain the probability distribution $p([\text{MASK}]|(X_{\text{prompt}}))$ of the candidate class, $y \in Y$ as:

$$p(y|X_{\text{prompt}}) = \sum_{w \in \mathcal{V}_y} p([\text{MASK}] = w|X_{\text{prompt}}) \tag{2}$$

where $w$ represents the $w^{th}$ label token of class $y$.

## 4   OUR APPROACH

### 4.1   MOTIVATION

It can be observed from the previous empirical findings (Gao et al. (2020); Scao & Rush (2021)) that an optimal prompt is necessary for the improvement of the pre-trained language models for the few-shot learners. Since templates with discrete tokens may be sub-optimal and are insufficient to represent a specific class[2], this study proposes **D**ifferenti**A**ble p**R**omp**T**, referred to as **DART**, which can reduce the requirement of prompt engineering in order to improve the applicability of the proposed method in various domains.

### 4.2   DIFFERENTIABLE TEMPLATE OPTIMIZATION

Since the language tokens are discrete variables, finding the optimal prompts with token searching is non-trivial and may easily fall into the local minima. To overcome these limitations, we utilize pseudo tokens to construct templates and then optimize them with backpropagation. Specifically, given the template, $\mathcal{T} = \{[\text{T}_{0:i}], [\text{MASK}], [\text{T}_{i+1:j}]\}$, which varies from the traditional discrete prompts, satisfying $[\text{T}_i] \in \mathcal{V}$ and map $\mathcal{T}$ into:

$$\{\mathbf{w}([\text{T}_{0:i}]), \mathbf{w}([\text{MASK}]), \mathbf{w}([\text{T}_{i+1:m}])\} \tag{3}$$

DART considers $[\text{T}_i]$ as pseudo tokens and maps the template as follows:

$$\{h_0, ..., h_i, \mathbf{w}([\text{MASK}]), h_{i+1}, ..., h_m\} \tag{4}$$

where $h_i (0 \le i \le j)$ are trainable parameters. Differentiable template optimization can obtain expressive templates beyond the original vocabulary $\mathcal{V}$. Lastly, the templates, $h_i$, are differentially optimized by:

$$\hat{h}_{0:m} = \arg\min_h \mathcal{L}(X_{\text{prompt}}, y) \tag{5}$$

Note that the values of the prompt embeddings, $h_i$, must be co-dependent with each other rather than independent. Unlike P-tuning (Liu et al. (2021c)), which utilizes a bidirectional LSTM, DART leverages an auxiliary fluency constraint objective to associate the prompt embeddings with each other, thus stimulating the model to focus on context representation learning.

### 4.3   DIFFERENTIABLE LABEL OPTIMIZATION

Prompt-based fine-tuning requires filling in one word, and the masked word prediction is mapped to a verbalizer, which produces a class (i.e., "Yes": True. "No": False). For each class $c \in Y$, the

---

[2]It is non-trivial to evaluate all options of templates and label tokens.

previous approaches such as LM-BFF (Gao et al. (2020)) estimate the conditional likelihood of the initial $\mathcal{L}$ on a pruned set $\mathcal{V}^c \subset \mathcal{V}$ of the top $k$ vocabulary words.

However, the brute-forcing label searching: (1) is computationally intensive and tedious because the $\mathcal{D}_{\text{dev}}$ is generally very large, requiring multiple rounds of evaluation. (2) has poor scalability with an increase in the class numbers (many classification datasets have more than 100 classes), the number of searches may be $k^C$ ($C$ represents the total number of classes), which is exponential and thus intractable. Additionally, the labels of classes contain rich, complex semantic knowledge, and one discrete token may be insufficient to represent this information.

Specifically, with the labels, $Y = \{Y_1, Y_2, .., Y_m\}$, different from the previous approach which converts the class type $Y_i$ into a variable number of label tokens $\{...,v_1,..,v_k,...\}$, DART maps the $Y_j$ to a continuous vocabulary space as follows:

$$\mathcal{M}(Y_j) = \{h_{m+j}\}, \tag{6}$$

where $m$ is the number of trainable embedding in template. To avoid optimizing any external parameters, $\{h_1, ..., h_m, .., h_{m+n}\}$ is replaced with unused tokens (e.g., [unused1] or special tokens in vocabulary) in $\mathcal{V}$ to generate $\mathcal{V}'$, as shown in Figure 1.

## 4.4 TRAINING OBJECTIVES

Since the pseudo tokens in the prompt template must be co-dependent with each other, we introduce an auxiliary fluency constraint training without optimizing any other parameters inspired by Liu et al. (2021c); Tam et al. (2021). Overall, there are two objectives: the class discrimination objective $\mathcal{L}_C$ and the fluency constraint objective $\mathcal{L}_F$.

**Class Discrimination Object** The class discrimination objective is the main objective that aims to classify the sentences. As shown in Figure 1, given $(X_{\text{in}}, \mathcal{T})$, we can generate $X_{\text{prompt}}$ as:

$$\mathcal{L}_C = \text{CE}(g(y|X_{\text{prompt}})). \tag{7}$$

where CE is the cross-entropy loss function, $\mathcal{L}_C$ represents the class discrimination loss.

**Fluency Constraint Object** To ensure the association among the template tokens and to maintain the ability of language understanding inherited from the PLMs, we leverage a fluency constraint object with the MLM. As shown in Figure 1, one token in the input sentence is randomly masked, and the masked language prediction is conducted. $x$ and $x'$ are the original and masked sequences, respectively. Let $x^m$ be the target token that has been masked out in $x'$, and $g(x^m|x', y)$ is maximized as follows[3]:

$$h(x^m|x', y) = \frac{\exp([\![f(x', y)]\!]_{x^m})}{\sum\limits_{v' \in \mathcal{V}'} \exp([\![f(x', y)]\!]_{v'})} \tag{8}$$

$$\mathcal{L}_F = \sum_{m \in M} \text{BCE}(h(x^m|x', y)). \tag{9}$$

By optimizing $\mathcal{L}_F$, the language model can obtain a better contextual representation with a rich association among the template tokens. We have the following training object:

$$\mathcal{L} = \mathcal{L}_C + \lambda \mathcal{L}_F, \tag{10}$$

where $\lambda$ is the hyper-parameter. Lastly, we introduce the overall optimization procedure of DART. To mitigate the instability of the few-shot fine-tuning, we jointly optimize templates and labels. Note that our approach can reuse the same transformer architecture (rather than additional LSTM) so that it enjoys the beauty of simplicity for prompt-tuning.

---

[3]We use the golden label $y$ rather than the [MASK] in the input of the fluency constraint object.

| Model | SST-2 (acc) | MR (acc) | CR (acc) | Subj (acc) | TREC (acc) |
|---|---|---|---|---|---|
| Majority[†] | 50.9 | 50.0 | 50.0 | 50.0 | 18.8 |
| Prompt-based zero-shot[‡] | 83.6 | 80.8 | 79.5 | 51.4 | 32.0 |
| "GPT-3" in-context learning | 84.8 (1.3) | 80.5 (1.7) | 87.4 (0.8) | 53.6 (1.0) | 26.2 (2.4) |
| Fine-tuning | 81.4 (3.8) | 76.9 (5.9) | 75.8 (3.2) | 90.8 (1.8) | 88.8 (2.1) |
| LM-BFF | 92.3 (1.0) | 85.5 (2.8) | 89.0 (1.4) | 91.2 (1.1) | 88.2 (2.0) |
| P-Tuning | 92.2 (0.4) | 86.7 (1.2) | 91.8 (1.1) | 90.3 (2.2) | 86.3 (4.5) |
| DART | **93.5 (0.5)** | **88.2 (1.0)** | **91.8 (0.5)** | 90.7 (1.4) | 87.1(3.8) |
| Fine-tuning (full)[†] | *95.0* | *90.8* | *89.4* | *97.0* | *97.4* |

| Model | MNLI (acc) | SNLI (acc) | QNLI (acc) | MRPC (F1) | QQP (F1) |
|---|---|---|---|---|---|
| Majority[†] | 32.7 | 33.8 | 49.5 | 81.2 | 0.0 |
| Prompt-based zero-shot[‡] | 50.8 | 49.5 | 50.8 | 61.9 | 49.7 |
| "GPT-3" in-context learning | 52.0 (0.7) | 47.1 (0.6) | 53.8 (0.4) | 45.7 (6.0) | 36.1 (5.2) |
| Fine-tuning | 45.8 (6.4) | 48.4 (4.8) | 60.2 (6.5) | 76.6 (2.5) | 60.7 (4.3) |
| LM-BFF | 68.3 (2.5) | 77.1 (2.1) | 68.3 (7.4) | 76.2 (2.3) | 67.0 (3.0) |
| P-Tuning | 61.5 (2.1) | 72.3 (3.0) | 64.3 (2.8) | 74.5 (7.6) | 65.6 (3.0) |
| DART | 67.5 (2.6) | 75.8 (1.6) | 66.7 (3.7) | **78.3 (4.5)** | **67.8 (3.2)** |
| Fine-tuning (full)[†] | *89.8* | *92.6* | *93.3* | *91.4* | *81.7* |

Table 1: Our main results with RoBERTa-large. †: the full training set is used. ‡: no training examples are used. Otherwise, we use $K = 16$ (# examples per class). We report mean (and standard deviation) performance over 5 different splits. Majority: majority class "GPT-3" in-context learning: using the in-context learning proposed in with RoBERTa-large (no parameter updates); LM-BFF: we report the performance in Gao et al. (2020). full: fine-tuning using full training set.

## 5 EXPERIMENTS

In this section, we detail the comprehensive experimental results conducted on classification tasks. The promising results demonstrate that our proposed DART substantially outperforms the conventional fine-tuning method, thus, making pre-trained language models better few-shot learners.

### 5.1 DATASET STATISTICS

We conduct a comprehensive study across 15 NLP tasks, which covers sentiment analysis, natural language inference, paraphrases, sentence similarity, relation extraction, and event extraction (We only report event argument extraction performance). The evaluation consisted of 10 popular sentence classification datasets (SST-2, MR, CR, Subj, TREC, MNLI, SNLI, QNLI, MRPC, QQP).To further evaluate the effectiveness of the proposed approach with complex label space, we conduct experiments on the relation extraction and event extraction datasets, including SemEval-2010 Task 8 (Hendrickx et al., 2010), TACRED-Revisit (Alt et al. (2020)), Wiki80[4] (Han et al., 2019), ChemProt (Kringelum et al., 2016), and ACE-2005[5].

### 5.2 SETTINGS

The proposed model is implemented using Pytorch (Paszke et al. (2019)). Our experiments are conducted with the same setting following LM-BFF ( Gao et al. (2020)), which measures the average performance with a fixed set of seeds, $S_{seed}$, across five different sampled $\mathcal{D}_{train}$ for each task. We utilize a grid search over multiple hyperparameters and select the best result as measured on $\mathcal{D}_{dev}$ for each set $\{\mathcal{D}_{train}^s, \mathcal{D}_{dev}\}, s \in S_{seed}$. We employ AdamW as the optimizer. We conduct experiments with a RoBERTa-large (Liu et al. (2019)) on classification tasks for a fair comparison with LM-BFF. We leverage an uncased BERT-large (Devlin et al. (2019)) for relation extraction datasets, except that we use SCIBERT (Beltagy et al. (2019)) for the ChemProt dataset. We follow Soares et al. (2019) and use special entity markers uniformly to highlight the entity mentions for relation extraction.

---

[4] https://github.com/thunlp/OpenNRE/
[5] https://catalog.ldc.upenn.edu/LDC2006T06

| Dataset | Model | $K = 8$ | $K = 16$ | $K = 32$ | *Full* |
|---|---|---|---|---|---|
| SemEval | Fine-tuning | 26.3 | 43.8 | 64.2 | 87.8 |
| | LM-BFF | 43.2 | 62.0 | 72.9 | 88.0 |
| | DART | **51.8** (+25.5) | **67.2** (+23.4) | **77.3** (+13.1) | **89.1** (+1.3) |
| TACRED-Revisit | Fine-tuning | 7.4 | 15.5 | 25.8 | 75.0 |
| | LM-BFF | 21.0 | 23.7 | 27.1 | 76.4 |
| | DART | **25.8** (+18.4) | **30.1** (+14.6) | **31.8** (+6.0) | **77.8** (+2.8) |
| WiKi80 | Fine-tuning | 46.3 | 60.3 | 70.0 | 87.5 |
| | LM-BFF | 66.5 | 73.5 | 78.1 | 86.2 |
| | DART | **68.5** (+22.2) | **75.2** (+14.9) | **79.4** (+9.4) | **88.1** (+0.6) |
| ChemProt | Fine-tuning | 30.2 | 41.5 | 52.5 | 79.5 |
| | LM-BFF | 55.0 | 56.1 | 60.0 | 79.1 |
| | DART | **57.2** (+27.0) | **60.8** (+19.3) | **63.1** (+10.6) | **81.0** (+1.5) |

Table 2: Results on RE dataset WiKi80 (accuracy), while other datasets (micro $F_1$). We use $K = 8, 16, 32$ (# examples per class). *Full* represents the full training set is used.

| Method | K=8 | K=16 | K=32 | Full |
|---|---|---|---|---|
| Conventional FT | 26.3 | 43.8 | 64.2 | 87.8 |
| DART | **51.8** | **67.2** | **77.3** | **89.1** |
| -fluency constraint object | 50.3 (-1.5) | 66.1 (-1.1) | 76.0 (-1.3) | 88.2 (-0.9) |
| -differentiable template | 49.8 (-2.0) | 66.3 (-0.9) | 76.2 (-1.1) | 88.4 (-0.7) |
| -differentiable label | 47.5 (-4.3) | 62.5 (-4.7) | 73.7 (-0.6) | 87.8 (-1.3) |

Table 3: Ablation of DART with different components on SemEval. (FT= Fine tuning)

## 5.3 MAIN RESULTS

As shown in Table 1, we observe that our approach obtains better performance than conventional fine-tuning and achieves comparable results with LM-BFF. Note that DART can reduce the prompt engineering without external models (e.g., T5 in LM-BFF) to generate templates that are readily easy to adapt to other datasets. DART can obtain 11.3% improvement with only 16 training samples per class on the MR dataset, comparable with LM-BFF, which leverages T5 to generate appropriate prompts. These results indicate that DART can better stimulate potential ability and makes the pre-trained language model a better few-shot learner. We also notice that DART yields better performance than P-tuning, which indicates that label optimization is beneficial.

For the classification tasks with the complex label space, as shown in Table 2 and Figure 2(a), we observe that DART outperforms the conventional fine-tuning approach as well as LM-BFF with a large margin on relation extraction and event extraction datasets in both the few-shot and fully supervised settings. The proposed approach achieves an improvement of 2.8% of the absolute performance on the TACRED-Revisit dataset with full supervision and yields 18.4% gains with only 8 training samples per class. These findings also indicate that more relevant templates and labels can be determined without expert intervention, making it possible to generalize the proposed approach to other domains. We attribute the significant improvements to the fact that, unlike the GLUE datasets containing small categories, in relation extraction and event extraction tasks, the datasets consist of a large number of classes with complex label space, making it more challenging to obtain suitable label tokens. Furthermore, we notice that the improvement decays slowly when $K$ becomes larger (i.e., from 8 to 32). Our approach is a simple yet effective fine-tuning paradigm that can reduce prompt engineering within the complex label space, thus, making it possible to be an appropriate plug-in for some SOTA models.

## 5.4 ABLATION STUDY

We conduct an ablation study to validate the effectiveness of the components in the proposed approach. We observe that DART exhibits a performance decay in the absence of any one of the modules, i.e.,

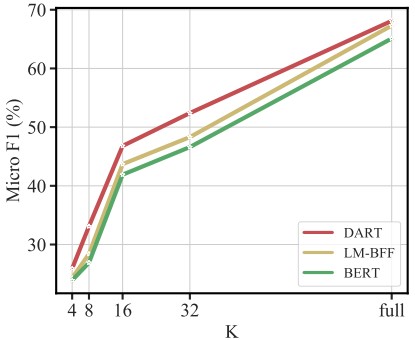 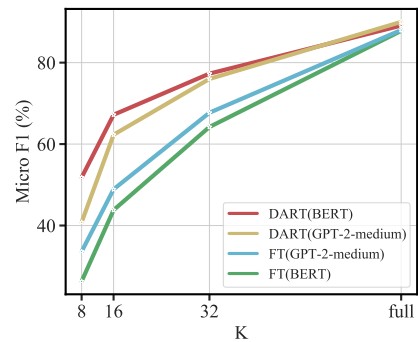

(a) Event extraction results on ACE-2005.     (b) BERT-large & GPT-2-medium results on Se-mEval.

Figure 2: (a) Few-shot results using the ACE-2005. We used K = 4, 8, 16, and 32 (# examples per class) with BERT. (FT= Fine-tuning) (b) BERT-large vs. GPT-2-medium results for the SemEval. Moreover, for lower K, our method consistently outperforms conventional fine-tuning.

fluency constraint object, differentiable template, or differentiable label, demonstrating that all the modules are advantageous. Furthermore, we notice that differentiable label optimization is more sensitive to performance and is highly beneficial for DART, especially for low-resource settings. Since the proposed approach is the first approach that utilizes the differentiable label optimization, these findings illustrate that a suitable label token is important.

## 5.5   ANALYSIS AND DISCUSSION

### CAN DART BE APPLIED TO OTHER PRE-TRAINED LMS?

To evaluate whether the proposed approach can be applied to other LMs, we conduct experiments using GPT-2-medium[6]. From Figure 2(b), we observe that DART with GPT-2-medium yields better performance than the conventional fine-tuning approach. Furthermore, we notice that DART with GPT-2-medium can achieve performance on par with BERT-large, as observed by Liu et al. (2021c), indicating that the potential of GPT-style architectures for natural language understanding has been underestimated.

### WHY DO DIFFERENTIABLE PROMPTS YIELD BETTER PERFORMANCE?

To further analyze why our differentiable prompts method yields better performance compared with prompts with fixed templates and label tokens, we visualize the representation of masked tokens in the CR dataset during different training steps (from left to right) as shown in Figure 3 (fixed) and 4 (differentiable), respectively. While both methods learn separable hidden states, differentiable prompts' representation is relatively more compact while the representation generated from fixed prompts is more scattered. This observation of differentiable prompts generating more discriminative representations than the fixed prompts method is supported by an indicator $R_D$, the ratio between average intra-class and average inter-class distance. We believe the main reason behind its better performance lies in the more discriminative representation of the differentiable method. More details can be found in Appendix A.6.

### WHAT EXACTLY IS OPTIMIZED PROMPT?

Since prompt templates and label tokens in the proposed approach are mapped as $\{h_1, ..., h_m, .., h_{m+n}\}$, we further analyze what exactly optimized label learned. We conduct a nearest-neighbor vocabulary embedding search to project the Top-3 optimized pseudo-label tokens in $\mathcal{V}$ to a readable natural

---

[6]We do not utilize the fluency constraint object in GPT-2-medium since the model is not pre-trained with MLM objective.

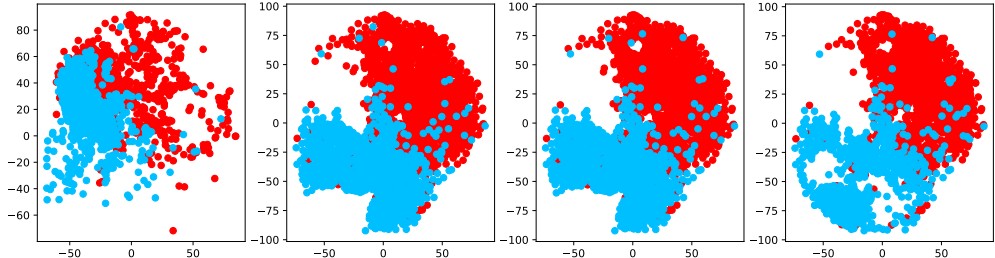

Figure 3: Visualization of masked tokens' representation in different training steps (with training 10, 30, 50, 70 steps from left to right) with *fixed prompts*.

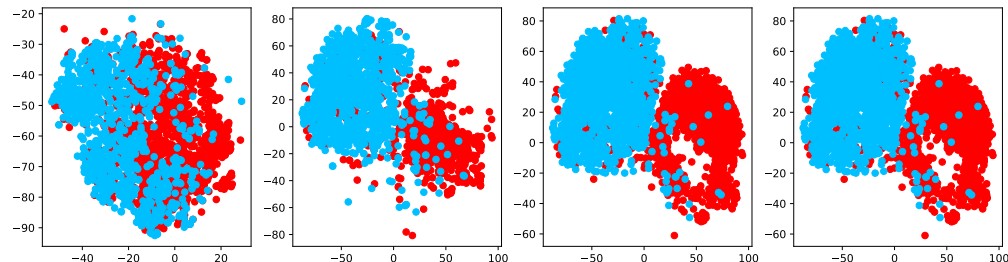

Figure 4: Visualization of masked tokens' representation in different training steps (with training 10, 30, 50, 70 steps from left to right) with *differentiable prompts*.

language.We use *t*-SNE (Van der Maaten & Hinton (2008)) with normalization to visualize labels on Wiki80 dataset. For example, "*military_branch*" refers to as red ⋆ in Figure 5 represents the relation type, which is learned by optimizing the pseudo label in the continuous space, and the "*volunteered*", "*corporal*" and "*buddies*", refers to as ● are the tokens closest to the label. This finding indicates that the differentiable method generates better semantic representation.

## DART v.s. Conventional Fine-tuning

The ability of DART to perform few-shot learning can be attributed to the label and being a true language understanding task, that once the model is capable of performing it correctly, it can easily apply this knowledge to other tasks that are framed as such. Superficially, (i) DART does not optimize any new parameters; however, conventional fine-tuning should learn an explicit classifier head over [CLS] embeddings, which may fail in the low-data regime. (ii) DART has the same task setting as large-scale language model pre-training.

Figure 5: A 3D visualization of several label representations learned in DART.

## 6 Conclusion and Future Work

This paper presents DART, a simple yet effective fine-tuning approach that improves the fast-shot learning pretrained language model. The proposed approach can produce satisfactory improvements in the few-shot scenarios when compared to the conventional fine-tuning approaches. The proposed method is also pluggable for other language models (e.g., BART) and can be extended to other tasks, such as intent detection and sentiment analysis. Intuitively, the results obtained in this study can be used to stimulate future research directions in the few-shot or lifelong learning for NLP.

## ACKNOWLEDGMENTS

We want to express gratitude to the anonymous reviewers for their hard work and kind comments. This work is funded by National Key R&D Program of China (Funding No.SQ2018YFC000004), NSFCU19B2027/NSFC91846204, Zhejiang Provincial Natural Science Foundation of China (No. LGG22F030011), Ningbo Natural Science Foundation (2021J190), and Yongjiang Talent Introduction Programme (2021A-156-G).

## REPRODUCIBILITY STATEMENT

Our code is available in `https://github.com/zjunlp/DART` for reproducibility. Hyper-parameters are provided in the Appendix A.1.

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

## A  APPENDIX

Our code is available in the supplementary materials for reproducibility. This section contains details about the training procedures and hyperparameters for each of the datasets. We utilize Pytorch (Paszke et al., 2019) to conduct experiments with 1 Nvidia 3090 GPUs. All optimizations are performed with the AdamW optimizer with a linear warmup of learning rate over the first 10% of gradient updates to a maximum value, then linear decay over the remainder of the training. Gradients are clipped if their norm exceeds 1.0, and weight decay on all non-bias parameters is set to 0.01. Early stopping is adopted to reduce over-fitting on the training set.

We follow LM-BFF (Gao et al., 2020) to measure the average performance of models trained on 5 different randomly sampled $\mathcal{D}_{train}$ and $\mathcal{D}_{dev}$ splits, and perform grid search for optimal hyper-parameter combinations on each split, including learning-rate, weight decay, and batch size.

For P-tuning (Liu et al., 2021c), due to the limit of search space, we do not set anchor tokens in prompt tokens.

For DART, we adopt joint optimization to acquire optimal prompts and fine-tune over global parameters. Note that we use base prompts as templates of pseudo tokens to accelerate convergence.

To compare fairly, we use RoBERTa-large (Liu et al., 2019) as pre-trained model for both DART and P-tuning framework, following LM-BFF (Gao et al., 2020). We also adopt the best discrete prompts together with label words in LM-BFF as base prompt settings for each framework, as stated below.

## A.1 Hyper-parameter Search Space of Our Method in Grid Search

### SST-2, MR, CR, Subj, TREC, QNLI, MRPC, QQP

The hyper-parameter search space is (the optimal set of parameters may vary across different tasks and data splits):

- learning rate [1e-5, **5e-5**, 1e-4, 2e-4]
- weight decay [0.0, **0.01**, 0.05, 0.10]
- number epochs [**20**,30]
- batch size: [4, **8**, 16, 24, 32]
- max seq length: 128
- gradient accumulation steps: [**1**, 2]

### MNLI, SNLI

The hyper-parameter search space is (the optimal set of parameters may vary across different tasks and data splits):

- learning rate [1e-5, **5e-5**, 1e-4, 2e-4]
- weight decay [**0.0**, 0.01, 0.05, 0.10]
- number epochs [**30**,40]
- batch size: [4, **8**, 16]
- max seq length: 256
- gradient accumulation steps: [1, **2**]

### TACRED-Revisit, WiKi80, SemEval

The hyper-parameter search space are:

- learning rate [3e-5,**5e-5**,1e-5,5e-6]
- number epochs [**20**,30]
- batch size: 48
- max seq length: 128
- gradient accumulation steps: 2

### ChemProt

The hyper-parameter search space are:

- learning rate [3e-5,**5e-5**,1e-5,5e-6]
- number epochs [**20**,30]
- batch size: 48
- max seq length: 256
- gradient accumulation steps: 4

### DialogRE

The hyper-parameter search space is (the optimal set of parameters may vary across different tasks and data splits):

- learning rate [1e-5, **5e-5**, 1e-4, 2e-4]

- weight decay [0.0, **0.10**]
- number epochs [20,30,**40**]
- batch size: [4, **8**]
- max seq length: 256
- gradient accumulation steps: [1, **2**]

## A.2 BASE PROMPT AND LABEL WORDS

**SST-2, MR, CR**

- prompt template(*length* = 3) ["*text*", "it", "was", "\<mask\>", "."]
- label words {"0": "terrible", "1": "great"}

**Subj**

- prompt template(*length* = 3) ["*text*", "This", "is", "\<mask\>", "."]
- label words {"0": "incorrect", "1": "correct"}

**TREC**

- prompt template(*length* = 1) ["\<mask\>", ":", "*text*"]
- label words {"0": "Description", "1":"Entity","2: "Expression","3": "Human","4": "Location","5":"Number"}

**MNLI, SNLI**

- prompt template(*length* = 2) ["*text$_a$*", "?", "\<mask\>", ",", "*text$_b$*"]
- label words {"contradiction": "No","entailment": "Yes", "neutral": "Maybe"}

**QNLI**

- prompt template(*length* = 2) ["*text$_a$*", "?", "\<mask\>", ",", "*text$_b$*"]
- label words {"not_entailment": "No","entailment": "Yes"}

**MRPC, QQP**

- prompt template(*length* = 2) ["*text$_a$*", "?", "\<mask\>", ",", "*text$_b$*"]
- label words {"0": "No", "1": "Yes"}

**TACRED-Revisit, WiKi80, SemEval,DialogRE**

- prompt template(*length* = 3) ["*text*", Entity1, "is", "the", "\<mask\>", "of", Entity2]
- label words {"country_of_origin", "participating_team", "participant_of",...}

## A.3 TEMPLATE LENGTH ANALYSIS

| Model | Accuracy |
|---|---|
| DART (*length* = 2) | 92.6 (0.6) |
| **DART**(*length* = 3) | **93.5 (0.5)** |
| DART (*length* = 5) | 91.2 (1.1) |
| DART (*length* = 10) | 90.6 (0.5) |
| Fine-tuning | 81.4 (3.8) |

Table 4: Few-shot performance on SST-2 task using templates with different length.

We define the length of a template as the number of tokens except for input sentence and <mask> token, and apply DART on templates with different length. The performance of a specific template length $l$ is derived by summarizing the averaging accuracy on each few-shot data splits, using template $T = t_1, t_2, ..., t_l$. From the Table 4, we observe that for the SST-2 task, the model whose template length is three yield best performance; however, the overall impact of template length is rather insignificant as models with different template length obtain relatively similar performance.

## A.4 Performance on Full Training Set

| Model | SST-2 (acc) | MR (acc) | CR (acc) | Subj (acc) | TREC (acc) |
|---|---|---|---|---|---|
| Fine-tuning | *95.0* | *90.8* | *89.4* | *97.0* | *97.4* |
| LM-BFF | 94.9 | **91.9** | 92.4 | 96.9 | 97.3 |
| DART | 94.6 | 91.3 | **93.8** | 96.6 | 95.6 |

| Model | MNLI (acc) | SNLI (acc) | QNLI (acc) | MRPC (F1) | QQP (F1) |
|---|---|---|---|---|---|
| Fine-tuning | *89.8* | *92.6* | *93.3* | *91.4* | *81.7* |
| LM-BFF | 89.6 | 90.3 | 92.8 | **91.7** | 86.4 |
| DART | 87.3 | 89.5 | 92.3 | 90.4 | **89.5** |

Table 5: Full training set results with RoBERTa-large. Fine-tuning: we reported same results as Gao et al. (2020). LM-BFF: we trained LM-BFF model (without demonstration) on full-training set.

We conduct experiments and report the performance of DART with full-sized training data of GLUE tasks. From Table 5, we notice that DART obtain better or comparable results compared with the standard fine-tuning and LM-BFF, indicating that prompt-based tuning methods benefit less from full-sized data.

## A.5 Performance with Constrained Label Tokens

We conduct a nearest neighbor vocabulary embedding search to project the best optimized differentialble label token to a readable natural token. Those tokens are chosen based on cosine-similarity between all tokens' embedding and the optimized differentialble label token of DART. We list them in descending order with similarity scores (i.e., the token 'great' is chosen as its cosine-similarity score with trained positive label embedding of DART is the highest among all tokens, and the token 'terrible' is the most similar token with the trained negative label embedding; the other tokens are selected and listed in descending order with similarity scores). From Table 6, we observe that the performance of fixed prompt models is related to the similarity score of the chosen label token and that the DART model learns more semantic representation for label tokens, thus, yield best performance.

| Label tokens | Accuracy |
|---|---|
| differentiable token (DART) | **91.8 (0.5)** |
| great/terrible | 91.5 (0.3) |
| fantastic/awful | 91.0 (0.6) |
| amazing/horrible | 90.2 (0.8) |
| good/bad | 89.6 (0.5) |

Table 6: Few-shot performance on CR task using constrained label tokens with DART.

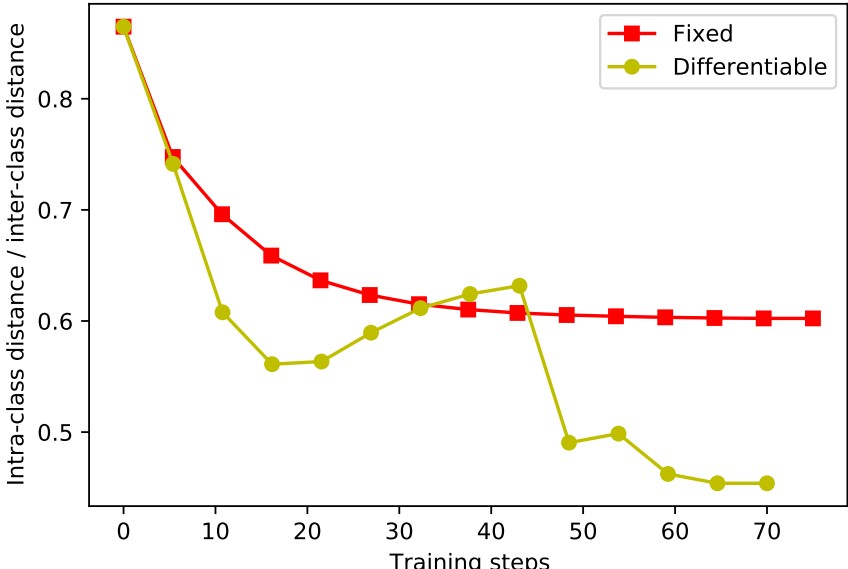

Figure 6: The $R_D$ ratio curve on dev set of CR task of fixed prompt and differentiable prompt during training.

## A.6 MORE EXPERIMENTS

We numeralize our observation on representation of masked token with a ratio between the average intra-class distance and average inter-class distance of hidden state vectors as $R_D = \frac{\bar{D}_{intra}}{\bar{D}_{inter}}$, where:

$$\bar{D}_{intra} = \frac{1}{C} \sum_{c=1}^{C} \bar{D}_{intra(c)} = \frac{1}{C} \sum_{c=1}^{C} \frac{1}{N_c} \sum_{i=1}^{N_c} \sum_{j=1}^{N_c} \text{distance}\left(H_c[i], H_c[j]\right);$$

$$\bar{D}_{inter} = \frac{1}{C(C-1)} \sum_{c_1=1}^{C} \sum_{c_2 \neq c_1} \bar{D}_{inter(c_1,c_2)} = \frac{1}{C(C-1)} \sum_{c_1=1}^{C} \sum_{c_2 \neq c_1} \sum_{i=1}^{N_{c_1}} \sum_{j=1}^{N_{c_2}} \text{distance}\left(H_{c_1}[i], H_{c_2}[j]\right);$$

(11)

where distance is the euclidean metric between two vectors, and $H_c[i]$ means the hidden state representation of masked token of $i$-th sample from class $c$. For discriminative representation, its average intra-class distance is low as data points within the same class tend to gather together, and its average inter-class distance is high as data points from different classes are separated, so its $R_D$ ratio should be close to 0.

As is shown in Figure 6, the $R_D$ ratio of the differentiable method grows lower than that of the fixed label method, which shows the hidden state representation trained in the differentiable method has better linear separability.

Note that in a masked language model, a linear transformation is performed on the hidden state representations, with a linear decoder sharing weights with the model's word embeddings serving as the final token classifier. Hence it is evident that better linear separability of the representations leads to better performance. In our case, the differentiable method yields better performance due to its better linear separability.

## A.7 LIMITATIONS

Our work may fail when the distribution of the task corpus varies from that of the pre-training corpus. For example, a general pre-trained language model may be fine-tuned with more training instances in a specific domain (e.g., medical domain). This issue can be addressed by intermediate training (Phang et al., 2018; Yin et al., 2020; Zhao et al., 2021), and will be analyzed in the future work. Besides, our work also shows an instability associated with hyper-parameters which is also observed by Dodge et al. (2020); Zhang et al. (2021); Perez et al. (2021) as volatility of few-shot learning

in NLP. Overall, however, we believe our work will inspire future work to few-shot settings with more practical applications to low-data settings, e.g., that involve low-resource languages or expert annotation.

## A.8 BROADER IMPACT

The pre-train-fine-tune approach has become the standard for natural language processing (NLP). However, supervised fine-tuning is still practically affected by labeled data. This study proposes a novel pluggable, extensible, and efficient approach named DifferntiAble pRompT (DART), which can convert small language models into better few-shot learners. We believe that our study makes a significant contribution to the literature because determining the appropriate prompts requires domain expertise, and handcrafting a high-performing prompt often requires impractically large validation sets, and these issues have been overcome with the use of the proposed method, which is model-agnostic, parameter-efficient. We experimentally verified our proposed approach on 13 standard NLP tasks, and it was seen to outperform several standard NLP platforms.

