# OpenReview forum: "Differentiable Prompt Makes Pre-trained Language Models Better Few-shot Learners"
_ICLR.cc/2022/Conference — ICLR 2022 Poster_

### Official Review · Reviewer_pzjK · 2021-11-02

**Correctness:** 4
**Technical Novelty And Significance:** 3
**Empirical Novelty And Significance:** 3
**Recommendation:** 8
**Confidence:** 3

**Main Review:**

Strengths:
 - The paper is generally well-written, with excellent motivation and empirical setup/analysis
 - The overall strategy of differentiable prompt optimized to maintain fluency is reasonable and novel.
 - The ablation experiments and optimized prompt analysis are insightful.

Weaknesses:
 - The notation/description of section 4 is not immediately intuitive. I had to read it a couple of times before I could fully follow the method. The authors could add more content to figure 1, which may resolve this issue.
 - The choice in deciding how many template tokens are used is unclear. An additional ablation experiment trying different number of tokens to optimize could be illuminating (even for just 1 dataset).
 - The ablation study would be better if specific numbers were provided.
 - I know GPT3 access is hard to get, but I wish experiments with k=8 prompts could be compared against

Post rebuttal: I think another pass for clarity over the paper would be good for the final version, but otherwise I'm happy with the paper updates and I'm happy to see the ablation numbers aren't too sensitive to token count. I have updated my score to reflect this.

**Summary Of The Paper:**

This paper proposes a new few-shot learning method for NLP problems by incorporating a simple,effective framework. This method is extensively validation and shows compelling performance.

**Summary Of The Review:**

This paper provides a simple, yet effective approach to the few-shot learning problem. While it has a couple minor issues, I think the broader community would find this paper interesting.

---

> ### Author Response · Authors · 2021-11-18
> **Response to Reviewer pzjK**
>
> Thank you for the detailed and constructive comments.
>
> We have carefully revised the notation/description of section 4 and also added content in Figure1.
>
> We have added an ablation study of different lengths of template tokens. From Table 5, we can observe that the number of template tokens impacts the model performance. In our previous experiments, we set the number of template tokens as hyper-parameters. Detailed settings can be found in Appendix.
>
> | Model                | Accuracy      |
> | -------------------- | ------------- |
> | DART($length=2$)     | 92.6(0.6)     |
> | **DART($length=3$)** | **93.5(0.5)** |
> | DART($length=5$)     | 91.2(1.1)     |
> | DART($length=10$)    | 90.6(0.5)     |
> | Fine-tuning          | 81.4(3.8)     |
>
> We have revised the paper and provided the specific numbers in the ablation study.
>
> Unfortunately, we do not have the GPT3 API and cannot conduct the experiments with k=8 prompts. We wish to add these experiments in the future.

---

### Official Review · Reviewer_LUWM · 2021-11-03

**Correctness:** 3
**Technical Novelty And Significance:** 3
**Empirical Novelty And Significance:** 3
**Recommendation:** 6
**Confidence:** 4

**Main Review:**

Strengths:

1) The paper proposes an interesting framework for few-shot fine-tuning without any prompt engineering. This approach can be easily used with other pre-trained models, which makes it a more generalizable framework.

2) The problem space is well defined (and also well contrasted with previous work), and the paper is clearly written and easy to follow.

Weaknesses:

1) Even though the results in Table 3 for relation extraction and event extraction datasets suggest that the proposed methods perform significantly better than LM-BFF, the results on the 10 popular datasets seems to be very close to LM-BFF. From the results, only two datasets (SST-2 and MR) seem to have significant improvement over LM-BFF. It would be great to further discuss these results in detail.


**Summary Of The Paper:**

The paper proposes a new approach called DART (Differentiable Prompt) which can perform few-shot fine-tuning without any prompt engineering (main difference w.r.t. previous works). This is achieved by optimizing the prompt template and the target label with backpropagation. Since the proposed approach doesn’t use any extra parameters, it can be easily used for any pre-trained language models. Further, the empirical evaluations suggest that it does better few-shot learning than previous works.

**Summary Of The Review:**

Overall, the approach is technically sound and is also very easy to apply to other pre-trained language models. The empirical results on the 10 popular datasets seem closer to LM-BFF and it’s hard to interpret if the approach empirically performs better than static/fixed prompts.  On the other hand, the proposed method performs much better on relation extraction and event extraction datasets, and it is not clearly discussed in the paper on these variations in the results.

---

> ### Author Response · Authors · 2021-11-18
> **Response to Reviewer LUWM**
>
> Thank you for the detailed and constructive comments.
>
> We have carefully revised the paper and analyzed the results. Specifically, in the few-shot setting on GLUE tasks, the proposed approach can obtain performance on a par with LM-BFF. On around half datasets, our method outperforms LM-BFF to a certain extent. However, the improvement is much more significant on relation extraction and event extraction tasks. We think this is due to the number of classes in those datasets. While most of the GLUE tasks contain only two or three categories, relation extraction and event extraction datasets contain more classes, making it even more challenging for previous models to obtain a suitable label token. Intuitively, our model can optimize the prompt template and the target labels with back-propagation and thus achieve better performance.

---

### Official Review · Reviewer_H1jf · 2021-11-03

**Correctness:** 3
**Technical Novelty And Significance:** 3
**Empirical Novelty And Significance:** 2
**Recommendation:** 6
**Confidence:** 4

**Details Of Ethics Concerns:**

-

**Main Review:**

Strengths of the paper:
- Clearly written and easy to follow.
- A simple, scalable, and effective approach of prompt tuning has been proposed.
- Importance of differentiable label has been well demonstrated.

Weaknesses of the paper:
- Missing information: Because the proposed Fluency Constraint Object is based on MLM, GPT-2-medium (in section 5.5 and figure 2-b) cannot use this objective.

Comments:
- In section 4.5, the authors said that their method requires no external parameters. But this claim could be controversial since adding new tokens (for prompts and labels) on the pre-trained language models' token embedding seems more efficient than pre-training language models that already have unused or special tokens in their token embedding. I know many PLMs already have unused tokens in practice, but I would like to treat this claim theoretically.
- For GLUE tasks, I'd like to know the performance of DART with full training set.
- Comparing with Prefix-tuning and WARP in experiments would be great. But this is not mandatory.
- In figure 3 and 4, training iteration or epoch should be presented.

**Summary Of The Paper:**

This paper proposes a new soft prompt tuning method called DART. The authors claim that training label representations for prompt tuning is important. With trainable label representations as well as prompt ones, auxiliary training task called fluency constraint object. Without the addition of a task-specific architecture, DART outperforms previous state-of-the-art prompt tuning in few-shot performance on 9/14 NLP tasks.

**Summary Of The Review:**

Recently, there has been heavy interests in prompt tuning such as P-tuning, Prefix-tuning, WARP due to its efficiency in use of large pre-trained language models for downstream tasks. The proposed DART has similar motivation. However, DART is more efficient than previous prompt tuning methods and is on par with them. Especially, focusing on differentiable label would be important observation for the community.

---

> ### Author Response · Authors · 2021-11-18
> **Response to Reviewer H1jf**
>
> Thank you for the detailed and constructive comments.
>
> Sorry for the missing details of GPT-2-medium. As GPT is an autoregressive language model, we directly append the prompt template with [MASK] at the end of the input sequence as "[cls] input [sep] template [sep]." For some specific PLMs, fluency constraint objects cannot be leveraged, but we have conducted the experiments and noticed that the proposed method still achieves better performance than baselines.
>
> We agree with the comment that "adding new tokens (for prompts and labels) on the pre-trained language models' token embedding seems more efficient." In our proposed model, the major difference is that it does not need external architectures such as LSTM; thus, it can be pluggable to any PLMs. We leverage a few parameters (unused tokens) in the language model to serve as the template and label tokens and optimize them via backpropagation. Thanks a lot for your constructive suggestions, and we will try to find some theoretical clues for this carefully in the future.
>
> We have revised the paper and added the performance of DART with the full training dataset in the Appendix.
>
> | Model       | SST-2(acc) | MR(acc)  | CR(acc)  | Subj(acc) | TREC(acc) |
> | ----------- | ---------- | -------- | -------- | --------- | --------- |
> | Fine-tuning | *95.0*     | *90.8*   | *89.4*   | *97.0*    | *97.4*    |
> | LM-BFF      | 94.9       | **91.9** | 92.4     | 96.9      | 97.3      |
> | DART        | 94.6       | 91.3     | **93.8** | 96.6      | 95.6      |
>
> | Model       | MNLI(acc) | SNLI(acc) | QNLI(acc) | MRPC(F1) | QQP(F1)  |
> | ----------- | --------- | --------- | --------- | -------- | -------- |
> | Fine-tuning | *89.8*    | *92.6*    | *93.3*    | *91.4*   | *81.7*   |
> | LM-BFF      | 89.6      | 90.3      | 92.8      | **91.7** | 86.4     |
> | DART        | 87.3      | 89.5      | 92.3      | 90.4     | **89.5** |
>
> This is an excellent suggestion, and we will try to conduct experiments to compare our model with prefix-tuning and WARP.
>
> We have carefully revised the paper and added steps in Figure 3 and 4.

---

### Official Review · Reviewer_EFFN · 2021-11-08

**Correctness:** 4
**Technical Novelty And Significance:** 4
**Empirical Novelty And Significance:** 3
**Recommendation:** 8
**Confidence:** 4

**Main Review:**

Strengths
- Very clearly written, the key ideas were well explained and simple.
- The results seemed convincing.
- As far as I can tell, all related works have been covered.

Weakness
- I am bit unconvinced about claiming that this work requires no external architecture in Table 1. There are new parameters that are getting trained/fine-tuned before we can do inference. The main difference I see between this and another approach like p-tuning is that the most of the existing architecture is getting reused vs in p-tuning there is explicitly an external LSTM.
- It would've been nice to see what happens if we constrain the additional tokens to be a part of the same vocabulary. This is more or less the approach taken in AutoPrompt, except with the current training methodology.

Questions
- I don't think I quite understood Sec 4.3, based on this section, it looks like there is an explicit embedding of output classes into an embedding? How are these learnt without fine-tuning? If they are learnt from scratch, then how is this different from regular fine-tuning?

- While the work claims that it reduces the need to fine-tune, I am seeing this work as a point on the spectrum between effort-to-tune vs zero-shot. Engineering prompts is truly zero-shot, P-tuning is somewhere in middle that it requires an external LSTM, whereas this approach requires retraining the big model (modulo all parameters except T1, T2.. held constant). Is this fair to say? How computationally easy is it to actually fine-tune these unused embeddings vs learning an external model

- How do we decide the number of unused tokens T1, T2..  to use?

- How much does this work translate to a completely novel task beyond classification? e.g. let's say constituency parsing or dependency parsing task, for which it is possible to engineer a novel prompt.







**Summary Of The Paper:**

This work reduces the need for prompt engineering for few-shot tasks by optimizing only the word-embeddings of unused tokens in the LLM. Results on multiple datasets showed reasonable results which IMO seems worth sharing to the broader audience. The exploration on what exactly are learned in the prompt was also interesting.

**Summary Of The Review:**

Clearly written paper with key insights nicely presented. I had some comments/questions on the key insights, but I don't think it should affect acceptance, unless I misunderstood.

---

> ### Author Response · Authors · 2021-11-18
> **Response to Reviewer EFFN**
>
> Thank you for the detailed and constructive comments.
>
> Our proposed model does not need external architectures such as LSTM; thus, it can be pluggable into any PLMs. We leverage a few parameters (unused tokens) in the language model to serve as the template and label tokens and optimize them via backpropagation. Thus, all optimized parameters are in the PLM, and we think we do not need external architectures in this way. We have carefully revised Section 4 and added contents in captions of Figure 1.
>
> Excellent suggestions. Concretely, our approach leverage learnable soft prompts while AutoPrompt utilizes discrete prompts. We have conducted experiments to constrain the additional tokens to be a part of the same vocabulary in the Appendix. We observe that the model can obtain a bit of performance decay compared with DART. We think this may further indicate that soft prompts can yield better performance than discrete prompts.
>
> | Label Tokens                | Accuracy      |
> | --------------------------- | ------------- |
> | differentiable tokens (DART) | **91.8(0.5)** |
> | great/terrible              | 91.5 (0.3)    |
> | fantastic/awful             | 91.0 (0.6)    |
> | amazing/horrible            | 90.2 (0.8)    |
> | good/bad                    | 89.6 (0.5)    |
>
> Sorry for some unclear parts in Sec 4.3. We have revised this section and also added contents in Figure 1. All the parameters in prompt templates, labels and other parameters in PLMs are optimized through fine-tuning.
>
> That is a fascinating question. We agree  with you that "this work as a point on the spectrum between effort-to-tune vs. zero-shot. Engineering prompts is truly zero-shot, P-tuning is somewhere in middle that it requires an external LSTM, whereas this approach requires retraining the big model". The primary motivation of our approach is the challenges for widespread classification tasks to obtain an optimized prompt template and target label token. Our approach obtains prompt templates and labels tokens via backpropagation without external architectures, unlike previous methods. Thus, our approach can be pluggable into lots of PLMs.
>
> We choose the number of unused tokens by evaluating the performance on the dev set. We also add an ablation study in the Appendix to further analyze the performance with different unused tokens.
>
> That is an excellent question. Recently, prompt tuning based approaches have obtained wonderful performance in classification tasks. However, for structure prediction tasks such as constituency parsing or dependency parsing, it is still non-trivial. We can treat them as natural language generation tasks and leverage approaches such as prefix-tuning. We will leave this as future works to extend our works to structure prediction.

---

> > ### Comment · Reviewer_EFFN · 2021-11-27
> > **Follow up**
> >
> > Thank you for the clear answers.
> >
> > I am still unconvinced with the author's response for Weakness#1 - "I am bit unconvinced about claiming that this work requires no external architecture in Table 1. There are new parameters that are getting trained/fine-tuned before we can do inference. The main difference I see between this and another approach like p-tuning is that the most of the existing architecture is getting reused vs in p-tuning there is explicitly an external LSTM.". This is similar to the concern that reviewer H1jf raised.
> >
> > As I wrote earlier, I still do think it is worth publishing this work despite this.

---

### Author Response · Authors · 2021-11-18
**Summary of Revisions**

We really appreciate all three reviewers for your valuable time and constructive comments.

We have uploaded a revised draft incorporating reviewer feedback. Below is a summary of the main changes:

- The experiments on different lengths of template (ununsed ) tokens in Table 5 (in Appendix). (R1 & R4)
- The experiments on GLUE tasks with full training datasets in Table 6 (in Appendix). (R2)
- The experiments with constrained label tokens in Table 7 (in Appendix). (R1)
- Section 4 and caption of Figure 1. (R1 & R4)
- Results analysis in Section 5.3. (R3)
- Ablation study in Section 5.3. (R4)

We sincerely hope our responses and revisions address all reviewers’ concerns.

We sincerely believe that these updates may help us better deliver the benefits of the proposed approach  to the ICLR community.

Thank you very much,

Authors.

---

### Public Comment · ~Carmen_Amo_Alonso1 · 2021-11-20
**More ablations are needed/fluency constraint is NOT a contribution of this work**

Hi
- the authors has considered adding a fluency constraint, in 8, this is first proposed by wellknown ADAPET paper [1], and this is not a contribution of this work. Unfortunately, this is not mentioned in the paper that fluency constraint is from ADAPET. This is mentioned this is inspired by them, but this is a direct usage of their work.
- second, as shown in ADAPET [1], adding this extra constraint by itself improves the results a lot, then this is not clear if the improvement of this work are coming from continuous labels or this extra condition, it would be great if authors could remove this constraint and provide the performance of their method so one can understand the improvement are coming from proposed continuous labels, or equivalently if they could add this constraint also for the baselines like LM-BFF and report their performance to provide a fair comparison.

thanks.

[1] Derek Tam, Rakesh R. Menon, Mohit Bansal, Shashank Srivastava, and Colin Raffel. Improving and
simplifying pattern exploiting training. CoRR, abs/2103.11955, 2021. URL https://arxiv.
org/abs/2103.11955.

---

> ### Author Response · Authors · 2021-11-21
> **Response to more ablations**
>
> Dear Carmen,
>
> Thanks a lot for your constructive comments.
>
> - When we are writing this paper, we notice ADAPET proposes label conditioning. Due to the fast development of prompt learning, different research teams may introduce some similar ideas. We have cited ADAPET in our earlier and revised version of the paper. Overall, our approach's major motivation and contribution is the differentiable prompt learning for few-shot NLP with optimized prompt templates and labels, which is different from ADAPET.
>
> - We have conducted ablation in Table 4. We observe that differentiable labels have the most impact on performance. Without the fluency constraint, the performance has a small drop in SemEval datasets. We have also conducted ablation studies in part of the datasets in GLUE. We find that differentiable label optimization is more sensitive to performance and is highly beneficial for DART, especially for low-resource settings. Still, the performance gains from the fluency constraint are relatively small.  As observed by [1] and  [2],  the verbalizer may be more important for prompt-based learning. We will try to report the results without the fluency constraint and LM-BFF with the fluency constraint. Our work and the well-known ADAPET are two different methodologies for prompt-based learning.
>
> We sincerely hope our reply can address your concerns.
>
> Thank you very much
>
> Authors.
>
> [1] Webson, Albert, and Ellie Pavlick. "Do Prompt-Based Models Really Understand the Meaning of their Prompts?." arXiv preprint arXiv:2109.01247 (2021).
>
> [2] Le Scao, Teven, and Alexander M. Rush. "How many data points are a prompt worth?." Proceedings of the 2021 Conference of the North American Chapter of the Association for Computational Linguistics: Human Language Technologies. 2021.

---

### Decision · Program_Chairs · 2022-01-20

**Decision:**

Accept (Poster)

**Comment:**

The paper presents a prompt learning method for few-shot learning in NLP.  In particular, they proposed DART, a new soft prompt tuning method, to optimize the label representations and template.

Overall, the paper is well-written and well-motivated. The proposed approach is interesting. The experiments were well justified and sufficient experimental analyses are provided. All reviewers support the paper.

There are a few remaining critics of the papers.

- The major one is the positioning of the paper raised by the reviewers. I agree with the reviewers that it is a bit misleading to emphasize the approach requires no external architecture. Although the approach can reuse the same transformer architecture (rather than additional LSTM) so that it enjoys the beauty of simplicity, it is still required additional parameters. I would suggest better clarifying this point in the final version.

- There is also a critic that the paper is related to ADAPET. However, the key ideas in this paper are sufficiently different from ADAPET. Also, ADAPET is published at EMNLP 21 after the paper is submitted, although it was in Arxiv earlier. It's fair to say this work is concurrent with ADAPET.